# Validation of Acoustic Wave Induced Traumatic Brain Injury in Rats

**DOI:** 10.3390/brainsci7060059

**Published:** 2017-06-02

**Authors:** Sean Berman, Toni L. Uhlendorf, David K. Mills, Elliot B. Lander, Mark H. Berman, Randy W. Cohen

**Affiliations:** 1School of Biological Sciences, Louisiana Tech University, Ruston, LA 71272, USA; sberman17@gmail.com (S.B.); dkmills@latech.edu (D.K.M.); 2Department of Biology, California State University Northridge, Northridge, CA 91330, USA; Toni.Uhlendorf@csun.edu; 3California Stem Cell Treatment Center, Rancho Mirage, CA 92270, USA; elliot@cellsurgicalnetwork.com (E.B.L.); mark@cellsurgicalnetwork.com (M.H.B.)

**Keywords:** traumatic brain injury (TBI), mild TBI, acoustic shockwave TBI, closed-head TBI, concussion

## Abstract

Background: This study looked to validate the acoustic wave technology of the Storz-D-Actor that inflicted a consistent closed-head, traumatic brain injury (TBI) in rats. We studied a range of single pulse pressures administered to the rats and observed the resulting decline in motor skills and memory. Histology was observed to measure and confirm the injury insult. Methods: Four different acoustic wave pressures were studied using a single pulse: 0, 3.4, 4.2 and 5.0 bar (*n* = 10 rats per treatment group). The pulse was administered to the left frontal cortex. Rotarod tests were used to monitor the rats’ motor skills while the water maze test was used to monitor memory deficits. The rats were then sacrificed ten days post-treatment for histological analysis of TBI infarct size. Results: The behavioral tests showed that acoustic wave technology administered an effective insult causing significant decreases in motor abilities and memory. Histology showed dose-dependent damage to the cortex infarct areas only. Conclusions: This study illustrates that the Storz D-Actor effectively induces a repeatable TBI infarct, avoiding the invasive procedure of a craniotomy often used in TBI research.

## 1. Introduction

Traumatic brain injury (TBI) is a severe injury affecting an estimated 3.8 million Americans and 10 million people globally each year with many more incidents going unreported [1,2]. TBI occurs when a blow to the head displaces the brain beyond the blood-brain barrier resulting in small lesions in the white matter in mild TBI, to more extreme subdural hematoma and damaged affected neurons in moderate and severe TBI [3]. When the brain is displaced beyond the cerebrospinal fluid it sits in, it collides into the bony skull and can cause blood vessels in the brain to rupture in moderate and severe TBI [4]. The disruption in the cerebral vasculature prevents adequate blood flow, depleting neurons of oxygen and essential nutrients. The neurons eventually die via apoptosis [5,6], and this form of neurodegeneration is directly associated with both the short-term symptomatic effects (poor motor coordination, severe headaches and complete loss of consciousness) and the long-term effects (memory loss, depression and suicidality) of TBI [7,8,9].

Military personnel in combat scenarios exposed to extreme overpressures associated with explosions experience TBI even though they may not come into direct contact with any solid objects. The pressure from these acoustic blasts emits forces equal to or greater than the physical force required for a mild TBI [7]. Blast TBI (bTBI) can often go undetected initially, but the typical short-term and long-term symptoms of TBI surface during the same timeline [7]. bTBI has been coined the “signature injury” of the recent wars in Iraq and Afghanistan with 22% of military personnel in these venues experiencing the adverse effects of bTBI [10,11,12]. Although the means of acquisition are different, the mechanisms and physiology of acoustic-produced TBI are similar to concussions that are often observed in athletics, automobile accidents, and everyday mishaps [7,13,14,15].

Experimental impact models have been developed to mimic human TBI in rodent models as well as other animals such as ferrets, cats, monkeys and swine [16,17,18,19,20]. Five standard impact models are currently used in research and include the fluid percussion impact (FPI), controlled cortical impact (CCI), penetrating ballistic-like brain injury (PBBI), Marmarou’s weight drop, and blast brain injury (bTBI) models. The open head models for TBI are FPI, CCI, and PBBI, all of which involve a craniotomy before administering the TBI. These methods may accurately model TBI specifically within the brain, but the surrounding biological chemistry may be altered due to craniotomy surgery and associated comprehensive damage. Furthermore, these open-head penetrating models may not be appropriate for administering regenerative stem cell therapy to treat TBI. For example, mesenchymal stem cells (MSCs) have shown the capacity to differentiate into neuronal cells. Major damage to the neighboring skull via craniotomy and destruction of surrounding tissue release a cascade of inflammatory cytokines, significantly reducing the ability of the MSCs to differentiate into neuronal cells [21].

Closed head TBI models are Marmarou’s weight drop and blast (bTBI) exposure. The Marmarou’s weight drop model is the most prevalent closed head technique used to study cellular and molecular responses to TBI. However, the precision of this technique depends on the animal’s neck musculature and may not be very reproducible with high mortality rates around 44% [22]. High-speed cameras also show that there may be a secondary, unintentional rebound injury in other regions of the brain after the initial impact of the weight drop [16]. The bTBI exposure models detonate a small explosive device or compressed air in a tube containing an animal in order to study the shockwave effect on the brain from peak pressure 154 to 340-kPa at different time intervals. However, the ensuing shockwave has been shown to cause injuries to air-filled organs not associated with TBI.

Brain injury severity is commonly classified as mild, moderate, and severe, which is based on the 15-point Glasgow Coma Scale (GCS) in humans [23]. The GCS scores from 13–15 is categorized as a mild TBI, 9–12 is moderate TBI, and 3–8 is severe TBI [24] There are six categories of symptoms that are currently used to clinically diagnose mTBI (mild traumatic brain injury): alteration, duration of loss of consciousness, Glasgow Coma Scale (30 min post injury with a score of 13–15), post-traumatic amnesia, focal, and brain imaging [23,24]. In animal models the primary injury of a mTBI cause the mechanical disruption of the tissue, the pathology of the secondary injury may ultimately cause further tissue damage and atrophy [25] as seen in the single mild TBI in a FPI animal study by Shultz, 2011 [26], and mild TBI in a CCI animal study by Yu, 2009 [27].

In our novel study, we present an acoustically produced TBI with similar brain injuries and concomitant behavioral abnormalities when compared to other animal models. Specifically, we illustrate that the acoustic wave technology of the Storz-D-Actor can deliver a precise controlled, closed-skull TBI that can be used in a myriad of other studies. We believe that there are two benefits for the use of this methodology. First, the acoustic shockwave model can accurately be calibrated to cause a range of brain damage in a specific region with reproducible results. Secondly, the acoustic TBI is not invasive and reduces the potentially secondary damage as the craniotomy surgeries in open-skull TBI.

## 2. Materials and Methods

### 2.1. Han-Wistar Rat

This experiment was performed using 60-day-old male rats (*n* = 40). Rats were taken from the Han-Wistar rat colony maintained at California State University Northridge. All protocols utilized in this study have been approved by both Louisiana Tech University’s Institutional Animal Care and Use Committee (IACUC) as well as California State University, Northridge’s IACUC.

### 2.2. Acoustic Wave TBI Induction Device: Storz-D-Actor

The Storz D-Actor (Storz Medical AG) is a device normally used in orthopedics, cardiology, urology, and aesthetic medicine to treat wide-ranging ailments utilizing its acoustic shockwave therapy. The device emits a precise, automated acoustic wave that creates a microtrauma in close proximity to wherever the handheld device is directed. The apparatus consists of a control unit, a pneumatic hand-held probe (tip diameter = 20 mm) and a pressurized air source. The control unit allows the user to select the frequency and pressure of the acoustic wave, ranging from 1–5 bar.

### 2.3. TBI Administration

Four different pressures were tested on four groups of Han-Wistar rats (*n* = 10 per treatment) in this experiment: 0 bar (control), 3.4 bar, 4.2 bar and 5.0 bar. To administer the brain injury, each rat was anesthetized briefly with 2.5% isoflurane and 95% O_2_ for 45 s. After anesthetization was determined, the rat’s head was stabilized and its ears pinned back using one hand while the other hand carefully positioned the hand-held probe of the Storz D-Actor on the top of the rat skull slightly posterior to the left eye, flush with the cranial epidermis (Figure 1). After the rat was secured, a single acoustic wave pulse was directed towards the left frontal motor cortex. The animal was then immediately placed on its back, and both the direction and time it took for the animal to roll over and right itself were recorded. Control rats were anesthetized and were treated similarly with a single, control pulse of 0 bar from the Storz-D-Actor probe.

### 2.4. Rotarod Test

Prior to TBI treatment, all animals underwent three, nonconsecutive days of training on a rotarod (Med Associates Inc., Fairfax, VT, USA) to measure the animal’s motor coordination skills both before and after treatment. The rotation of the rod slowly increased (4–40 rpm) over the course of the 300 s trial. Each day, their latency times (time spent on the rotating b) were recorded for three trials and averaged. By the last day of training, all animals were able to stay on the rotarod for all three, 300 s trials. After the TBI was administered on day 0, the rats were retested on the rotarod on days 1, 4, 7 and 10. At this point, we extended time on the rotarod to three, 360 s trials as, the control rats’ performances improved with the additional training. Latency times (s) for the control and treated rats to remain on the accelerating rotarod were recorded.

### 2.5. Water Maze Test

To test for memory impairment, we utilized a round black plastic tub (1 m diameter) divided into four quadrants was used as our water maze. The tub was filled up to 20 cm with water (approximately 19 °C), which was made opaque by using dehydrated milk powder. Above the water line, each quadrant was marked with a unique symbol using white tape. The clear platform 5 cm in diameter sat in the same location (2.5 cm below the waterline in the 3rd quadrant) for all sessions. The rats were placed in the water facing away from the platform in the 1st quadrant. The time (swim latency in seconds) for the rat to turn around, swim to the hidden platform and stand on top of it for a full second was recorded. Each water maze test was conducted after completion of rotarod testing. Like the rotarod test, the animals underwent three, nonconsecutive days of training in the water maze prior to TBI treatment. Three trials were completed each day with five minutes of rest between each trial. After TBI induction, animals were retested on days 1, 4, 7 and 10. Their swim latency times were recorded for three, nonconsecutive trials and averaged.

### 2.6. Histological Analysis

Ten days post-TBI administration (and after the final behavioral assays were completed), all animals were anesthetized with 400 mg/kg chloral hydrate (IP). The animals were then transcardially perfused and fixed with 4% paraformaldehyde dissolved in 0.1 M phosphate buffered saline (PBS). Their brains were removed and post-fixed in 4% paraformaldehyde/0.1 M PBS for 48 h at 4 °C. The fixed brains were then placed in a solution of 20% sucrose in paraformaldehyde/0.1 M PBS for an additional 48 h (also at 4 °C) prior to sectioning. The cerebrum was sliced on a cryostat on the coronal plane 25 μm thick at the same stereological level for all animals. The tissue was arranged on a glass slide for drying and was hydrated in 100%, 95% and then 70% ethanol baths, respectively for 2 min each before being stained in cresyl violet for 45 s. The tissue was washed in distilled water and distained in acetic formalin for 5 min. The tissue was then dehydrated in 95%, 100% and 100% ethanol solutions followed by xylene for 2 min each. To preserve the tissue, Permount was added and sealed with a cover slip. The slides were examined under microscopy for any damaged tissue, cell death and any disruption or skew from the acoustic wave induced TBI.

In addition, histological slides from the 3.4, 4.2 and 5.0 bar treatment groups were analyzed for degree of injury. The slides used for the infarct measurements were sampled from the same stereologic region of the injured frontal cortex (approximately +3.0 anterior to bregma). Quantification of the TBI infarct area was determined by the disruption of condensed dark cells. These dead cells can be easily visualized by their shrunken, condensed, dark staining nuclei and are located along the periphery of the infarct area [28]. The damaged cortical area surrounding the TBI was traced by following the boundary of these peripheral dark cells. The infarct area was then quantified using SketchandCalc^TM^ (iCalc, Palm Cost, Florida, USA) to measure the region of damage (mm^2^).

### 2.7. Statistics

Statistical analyses were performed on the experimental groups. Repeated measure ANOVA was performed on the data collected to complete the rotarod and water maze tests. Correlation (*R*^2^) values were obtained to determine if there was a correlation between the intensity of the TBI (righting times) and subsequent reductions in motor skills and memory testing. Finally, an ANOVA was used to examine the histology data to test for significance. All values shown in each figure were means ± standard errors.

## 3. Results

### 3.1. Immediate Post-TBI Observations

Immediately after acoustic TBI was administered to the left frontal cortex, every rat that received a 3.4, 4.2 or 5.0 bar TBI rolled and spun to its contralateral side (right) between one and four times before righting itself fully. In contrast, rats in the control group (0 bar) displayed the ability to right themselves towards the right or left side equally, often rolling back and forth. In addition, rats in the TBI test groups visually appeared to be breathing much heavier after the administration of the TBI. Breathing had slowed while chest palpitations became much more pronounced. All rats in the control group visually showed normal breathing rates after application of isoflurane anesthesia.

In addition, the amount of time the animals spent unconscious was quite variable. Overall, animals receiving TBI took longer to right than those just receiving anesthesia and sham (0 bar) TBI. Control animals receiving anesthesia righted themselves on average in 11.6 s compared to the 3.4 bar, 4.2 bar and 5.0 bar groups which righted on average 63.9 s, 31.5 s and 39.3 s, respectively. Interestingly, of the animals that received TBI, those that received a 3.4 bar bTBI consistently righted itself the slowest, almost twice that of the other two groups.

### 3.2. Rotarod Test to Assess Motor Skills

The animals in each TBI group were tested on the rotarod for deterioration of their motor skills post-treatment. Their results were averaged and displayed in Figure 2. All animals in the control group were able to exceed their training level performances and were allowed to continue for an additional 60 s (up to 360 s in total time on the rotarod) throughout the course of the 10-day post-treatment study. However, each of the TBI test groups showed decreased performances in motor skills ability over the 10-day, post-TBI assay. Specifically, the 4.2 bar test group showed the most decreased ability even one day after administration of TBI, decreasing by about 23% compared to controls. The 5.0 bar test group showed a more gradual decline in motor skills ability from day 1 to day 7, declining by 23% on day 7, before showing slight improvements on day 10, down only 17.5%. Finally, the 3.4 bar test group showed initial declines of about 9% on day 1, fell further to 18% on day 4 before trending up to 4% deficits by day 7. A repeated measures ANOVA showed statistical differences (*F* = 5.01; *p* < 0.005) for the rotarod test between control and treatment groups. However, there were no statistical differences among the TBI experimental groups regardless of wave intensity.

We also correlated the amount of time an animal spent unconscious with motor skills performance on the rotarod test. We compared unconsciousness time with their worst performance (the shortest latency time they spent on the rotarod) independent of which day this occurred. There were no significant correlation effects observed with *R*^2^ values for the 3.4 bar, 4.2 bar and 5.0 bar were 0.031, 0.009 and 0.001, respectively. The *R*^2^ value for the control group was 0.0517, also non-significant.

### 3.3. Water Maze Test to Assess Memory Impairments

To test for memory dysfunction as a symptom of TBI, animals in each treatment group (*n* = 10 for each treatment) completed the water maze assay on pretreatment day (day 0) and then post-treatment days 1, 4, 7 and 10. Their results were averaged and displayed in Figure 3. The control group showed the most improvement in finding the hidden platform compared to the TBI experimental groups as their times improved throughout the experiment. The 3.4 bar group regressed by 21% from day 0 to day 1, but then saw increases in swim latency times regressing 61% from day 0 to day 4 and regressing 84% by day 7, before settling down 42.5% on day 10. The 4.2 bar test group also had significantly slower latency times, down 51%, one day after TBI. The 4.2 bar group recovered slightly, down only 40% by day 7; yet, on day 10, the 4.2 bar rats took even more time (66% slower) to find the hidden platform. Likewise, the 5.0 bar test group showed a slight 20% increase in swim time to find the platform on day 1. On day 4, their swim time slowed over 8 s, down 115% from baseline. From day 7 to day 10, swim times reversed similar to the 4.2 bar group, and were 33.7% slower than pretreatment values. Repeated measures ANOVA (*F* = 2.80; *p* < 0.05) showed significant differences between control and treated TBI groups. Similar to the rotarod assay, there were no statistical differences among the TBI experimental groups in the water maze assay regardless of wave intensity.

As with the rotarod assay, we examined the correlation with time (s) an animal spent unconscious and their worst water maze performance (the longest time taken to find the hidden platform) on the water maze test. Similar to the rotarod performance, there were no significant effects observed with the *R*^2^ values for the 3.4 bar, 4.2 bar and 5.0 bar respectively were 0.0002, 0.0007 and 0.1225. The *R*^2^ value for the control group was 0.070 and was non-significant.

### 3.4. Histological Assay

Microphotographs of cresyl violet stained tissue from 3.4 bar TBI treatment (Figure 4A,C) show indentation and fragmentation of the cortex exactly where the acoustic wave was administered to two separate rats. The opposing panels (Figure 4B,D) show intact, unaffected tissue in the contralateral cortex at the same stereologic orientation (approximately +3.0 anterior to bregma). In addition, the cells in Figure 4A,C appeared to be more sparsely distributed in the damaged region, suggesting neurodegeneration and disruption of neural pathways. Similar histological profiles were observed with 4.2 bar sections.

Figure 5A shows the TBI-targeted left frontal cortex 10 days after 5.0 bar acoustic wave treatment. Note the severely damaged cortex compared to 3.4 bar treatment cortex (see Figure 4A,C). While the histology from the 3.4 bar treatment revealed moderate impairment of the dura mater and the arachnoid layer, the 5.0 bar treatments showed significantly more damage to post-TBI tissue that ranges well beyond the pia mater. The intact contralateral cortex showed no damage (Figure 5B). Also, no additional damage was detected in any other brain region of the TBI-treated rats, including the hippocampus.

Figure 5C,D are higher magnification photomicrographs taken from the highlighted regions of Figure 5A (Figure 5C upper region, Figure 5D lower region of Figure 5A). Figure 5C shows the damage taken from the damaged impact zone at the top of the infarct. Very few intact pyramidal neurons remain in this region. Notice the lightly stained granulated cells that appear to be leukocytes infiltrating from capillaries due to disruption of the blood-brain barrier [29]. Figure 5D shows the continuation and persistence of damaged cells throughout the parenchyma of the infarct region with the arrows showing granulated cells. These cells are most likely macrophages that have phagocytized blood products or other injured cells [29]. Large reparative astrocytes and condensed dark cells can be seen in Figure 5D as well.

A side-by-side comparison of all four treatments is shown in Figure 6. An intact, uninjured cortex revealed no detectable damage (Figure 6A). Slight damage of the outer cortex that does not penetrate beyond the pia mater is seen in the 3.4 bar treatment (Figure 6B). The injury from the 4.2 bar treatment is much more significant than that of the 3.4 bar, penetrating beyond the pia mater and reaching cortical neurons (Figure 6C). The depth of the 4.2 bar injury can be compared to that of the 5.0 bar injury (Figure 6D). The 5.0 bar treatment is much more pervasive and penetrates even deeper into the cortex. The clearer space within the marked region (Figure 6D) clearly shows neuronal cell death.

Finally, the depth of TBI-induced injury was analyzed from the tissue slices from the 3.4, 4.2 and 5.0 bar treatment groups. The damaged cortical area from each animal was traced and outlined (see Figure 6), and a mean injured area was determined (Figure 7). The 3.4 bar treatment group had an average injury area of 1.07 mm^2^. The 4.2 bar treatment group had an average injury area of 4.36 mm^2^ nearly four times greater than damage caused by the 3.4 bar treatment. Finally, the 5.0 bar treatment group had an average injury area of 8.15 mm^2^ which was nearly eight times greater than the 3.4 bar and nearly two times greater than the 4.2 bar injury. An ANOVA showed statistical differences among the treatment groups (*F* = 134.90; *p* < 0.001) in a dose-dependent manner.

## 4. Discussion

The novel acoustic wave TBI model presented in this study fulfills the criteria for behavioral, neuropathological, and histological outcomes involved in mild traumatic brain injury (TBI) similar to damages observed in other animal TBI models. In constructing the validity of acoustic wave induced TBI, we have shown this technique to be highly reproducible in the areas of behavioral response, motor skills, memory analysis and tissue damage. The acoustic wave TBI is a closed skull model making it clinically advantageous to the development of potentially innovative treatments, including possible stem cell therapies.

All acoustic wave TBI treatments were administered to the left frontal motor cortex the area of the brain responsible for voluntary movement as well as memory and decision-making. Rats that received isoflurane anesthesia and sham-TBI treatment moved around from left to right equally upon waking and righting. Conversely, 100% of the experimental rats that received treatment regardless of the amplitude always rolled to their right (contralateral) side upon righting, confirming that the acoustic wave generated from the Storz D-Actor consistently administered the same severity of injury targeted to the same region of the cortex. Our results were ultimately confirmed by histological analysis that showed that the acoustic-wave TBI technique was highly reproducible.

Besides a consistent righting response, we also analyzed the treated animals for motor skills and memory disturbances caused by acoustic wave TBI. Figure 2 shows the variation between the control and the TBI test groups in the rotarod assay. Post-TBI analysis throughout the duration of the 10 days of testing showed that the control and TBI test groups differed by a significant 20% reduction in the rotarod latencies (*p* < 0.01). Our study clearly indicated that the acoustic wave induced TBI caused substantial damage to the motor cortex at least over the ten-day test period.

Like the rotarod motor skills testing, we saw similar significant trends in the water maze assay. First, a rapid initial decline in performance was seen followed by slight improvements, then declining again by the final day of testing (Figure 3). Specifically, we observed a significant decline in memory recall by rats that received TBI. The administration of TBI caused the 3.4 bar, 4.2 bar and 5.0 bar groups to perform 37.2%, 38.1% and 41.6% slower, respectively. In contrast, the control group improved their times by 39.6% by the final day of testing.

It has been suggested that the amount of time an animal spends unconscious as a result of TBI directly affects post-TBI memory and motor performance [30]. Yet in our correlation study, we observed no significant long-term effects of the anesthesia, suggesting that the amount of time the rats remained unconscious due to the combined consequences of TBI and anesthesia had no relationship to the animal’s post-TBI memory or motor skills performance.

TBI is initially a vascular injury in nature [28] as trauma causes damage to blood vessels that supply neural tissue with nutrients. The rupturing of the blood–brain barrier leads to inflammation, which is responsible for the headaches and dizziness in human patients. We speculate that these initial effects caused the observed day 1 declines in rotarod and water maze performances compared to both their pre-TBI (day 0). In contrast, Figure 2 and Figure 3 show the animals improving albeit slightly after the initial injury insult on and post-TBI day 4 and 10 results. We hypothesize that this post-TBI improvement is due to the quick healing of the initial vascular injury, mitigating the headaches and dizziness often associated with TBI. While the initial vascular injury caused swelling [31], neurons were able to continue with a compromised nutrient supply that may have caused these cells to eventually die, resulting in the observed reduced motor skills and memory performances. This hypothesis was correlated by our histological analyses.

Indentation and fragmentation of the surface of the frontal cortex was observed where the acoustic waves were administered and focused (Figure 4 and Figure 5). The severity of the acoustic TBI has been shown to correlate with the acoustic wave intensity (Figure 6 and Figure 7). Figure 5C shows injured compacted neural cells, many of which appear condensed and dark. Figure 5D shows the perimeter of the infarct area visualized by the condensed dark cells. The impact from the 5.0 bar treatment severely ruptured blood vessels in the dura mater and the subarachnoid space before penetrating down past the pia mater, evidenced by granular phagocytized blood cells or the remnants of neurons (dark cells), also seen in Figure 5C. These images (Figure 5C) show the presence of phagocytic leukocytes near the impact zone, illustrating that the force from the TBI fractured blood vessels and caused the blood–brain barrier to rupture at least temporarily. Mukherjee et al. [29] suggested that these observed macrophages have entered into the TBI infarct to phagocytize the newly entered blood cells which elicits a yellowish stain from ingested red blood cells (these cells do stain yellow in the original color photomicrographs). The upregulation of expressed cytokines may also attract the infiltration of neutrophils to aid in cellular repair but may also lead to further degradation of the blood–brain barrier [5].

The pervasive nature of the acoustic wave is best observed beyond the initial point of impact. Figure 5A,B as well as Figure 5C,D show the difference in the neuronal cells on the impacted side (A, C, and D) and non-impacted side (B) 10 days after TBI. The cells from the damaged, left cortex are enlarged, and appear to be more disorganized and skewed than comparable cells in the right hemisphere. There are many hypertrophic astrocytes that have moved in to help repair the vascular brain injury, a sign of the increased severity of the injury [32].

The increasing size of the TBI is dependent on acoustic wave intensity and can be observed in Figure 6 and Figure 7, implicating the scalable nature of the Storz-D-Actor acoustic wave, produce a mild TBI (Figure 6B) or a more intense moderate TBI (Figure 6C,D). Although not part of this research, the Storz-D-Actor could conceivably produce a more severe TBI with repeated applications, resulting in much greater cell death. While disruption and fragmentation of the dura mater seemed consistent throughout the test groups, wave intensity correlated to both the size and depth of TBI (Figure 6 and Figure 7). Subsequent measurements of TBI infarct area (Figure 7) corroborate the visual evidence in the photomicrographs. TBI injury causes pyknosis, defined as when neurons condense and become darkly stained cells [33,34]. Other studies have observed similar results with dark neurons expressing apoptotic proteins Bax, BCl2 and cleaved caspase-3 [28]. Other experiments have identified the presence of jellyfish microglia that are activated after TBI insult and are part of the inflammatory process responsible for neuroprotection and reparative are cells, aiding in neuronal recovery [35].

The histological and behavioral results from our dose-response experiment advocate the applicability of an acoustic wave-induced TBI. Beyond invasive TBI models that utilize craniotomies, current closed-skull TBI models require subsequent blunt force trauma to the brain. Fragmentation and disruption of the outer cortex beyond the pia mater were observed in these studies. Such traumas have been shown to damage and kill neuronal tissue. Similarly, the acoustic wave generated by the Storz D-Actor illustrates identical histological effects (dying neurons, macrophages, astrocytes, etc.) previously seen in other TBI models [33,34,35]. In addition, TBI models consistently exhibited many damaged neurons manifesting as dark cells [33,34,35]. As the intensity of the acoustic wave increased from 3.4 bar to 5.0 bar, so did the number of observable dark neurons, consistent with the other blast intensity [33] and controlled cortical impact TBI studies [28]. Finally, sheering of the cortex tissue and opening the blood–brain barrier, appeared to recruit macrophages that have phagocytized blood products to the region, which has been also seen in fluid percussion injury models [18].

The behavioral results observed in the rotarod test were also comparable to previous studies that showed behavioral delinquencies immediately post-TBI and often depicted delayed setbacks in return to testing baselines one-week post-TBI [9,36]. In addition, TBI rats that performed the water maze memory test displayed delays of 5–15 s in finding a hidden platform compared to control rats [37]. Different TBI methods also revealed significantly increased swim latency times following TBI, taking a period of days to weeks to fully recover to baseline [38]. Although the hippocampus was not directly impacted during this study, it has been suggested that damage to the prefrontal cortex can result in increased swim latency [38]. Also, sleep patterns essential to memory formation and retention can be affected by a TBI, possibly causing increased swim latencies [39].

Our study looked to validate a practical and realistic TBI model that could be easily used on animal models in a laboratory setting. Many researchers have used invasive craniotomies in inducing multiple kinds of brain injuries. The craniotomy methods alone can induce various unintended consequences and make it difficult to establish proper controls in such studies [40]. By using the acoustic wave technology of the Storz-D-Actor, we showed conclusively that a TBI could be established in an animal model that avoided damage to the bony skull itself, and instead focused the insult energy on a specific region of the frontal cortex. Ultimately, we believe that the Storz-D-Actor acoustic model more accurately represents many of the TBI injuries, including the bTBI that military personnel receive at war, a TBI caused by overpressure from a nearby explosion in which no skull fracture is observed. Since the physiology of a bTBI and regular TBI are similar in terms of symptoms and histology, we also anticipate that this model can be used to study TBI sustained in other concussion models, including vehicle collisions, falls, and athletics (football, boxing, soccer, and hockey).

## Figures and Tables

**Figure 1 brainsci-07-00059-f001:**
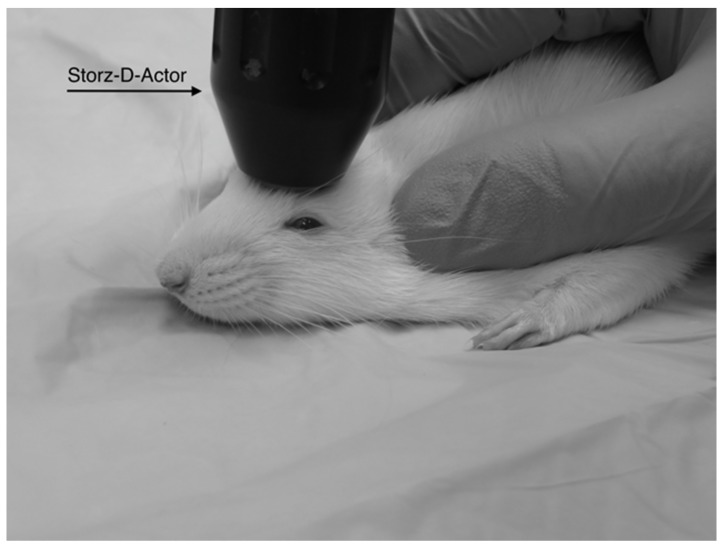
A diagram shows the placement of the Storz-D-Actor (D20-S 20 mm radial head applicator) being applied to a rat brain. The Storz-D-Actor is held at 90° to the surface as the energy is directed straight down through the left frontal cortex (approximately +3.0 anterior to bregma).

**Figure 2 brainsci-07-00059-f002:**
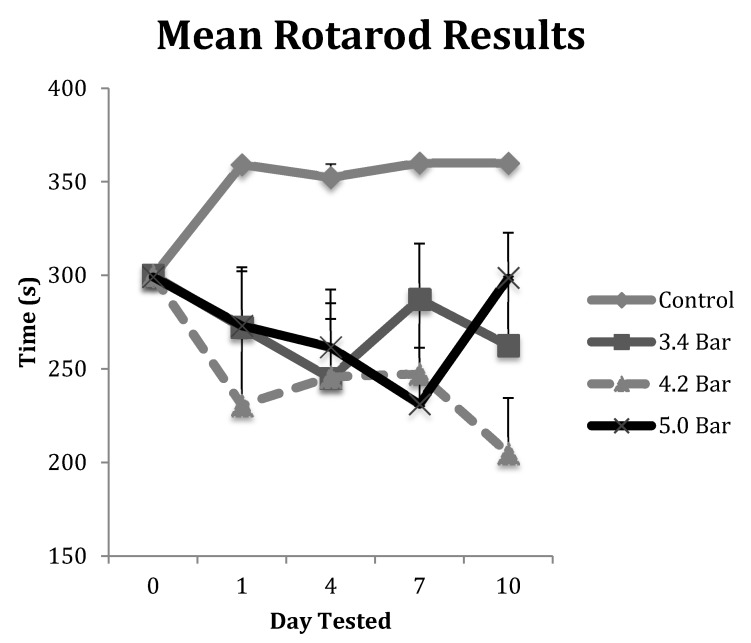
The rotarod test was performed to assess the decline in motor coordination of Han-Wistar rats after traumatic brain injury (TBI) treatment. The graph displays the mean time spent on the rotating bar (s) of control and TBI rats from Day 0 (prior to TBI treatment) to 10 days post-treatment. Treatments included control (0 bar), and experimental animals 3.4 bar, 4.2 bar and 5.0 bar (*n* = 10 for all treatments). The graph shows a significant decline in motor skills of TBI treated animals compared to controls (Repeated Measures ANOVA; *F* = 5.01; *p* < 0.005). All values are means ± standard error.

**Figure 3 brainsci-07-00059-f003:**
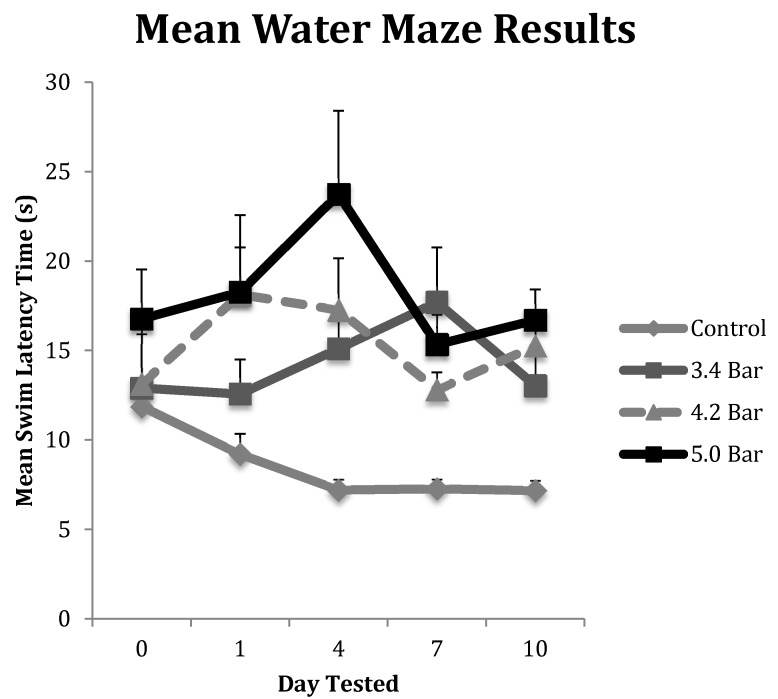
The water maze was used to assess the spatial memory of Han-Wistar rats post-treatment compared to the control group. The graph displays the mean swim latency times of control and TBI rats from Day 0 (prior to TBI treatment) to 10 days post-treatment. Acoustic wave treatments included a control (0 bar), and experimental 3.4 bar, 4.2 bar and 5.0 bar (*n* = 10 for all treatments). The results indicate a significant increase in timed swim latency of TBI-treated animals compared to untreated controls (Repeated Measures ANOVA; *F* = 2.80; *p* < 0.05). All swim latency values are means ± standard error.

**Figure 4 brainsci-07-00059-f004:**
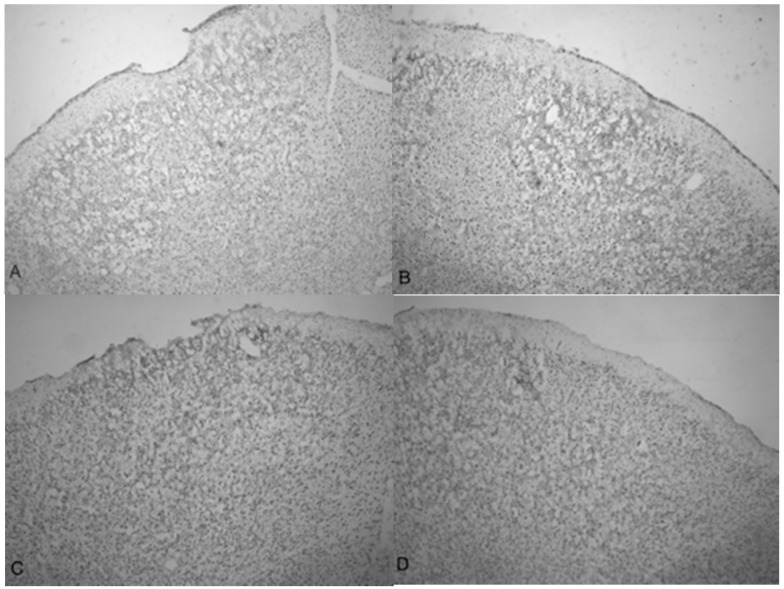
Microphotographs show the tissue from the frontal cortex visualized with cresyl violet from 60 day-old Han-Wistar rats, 10 days post-injury. Images indicate the damage from the acoustic wave 3.4 bar TBI targeted left frontal cortex as shown in the left panels (**A**,**C**) and compared with the contralateral, unaffected right frontal cortex (**B**,**D**) from the matching animals. Panels (**A**,**C**) show indentation and fragmentation of the cortex where TBI was administered while panels (**B**,**D**) show intact, unaffected tissue. In addition, the cells in A and C are more sparsely detected, suggesting neurodegeneration and disruption of neural pathways in these animals. All photomicrographs were taken at 40× magnification.

**Figure 5 brainsci-07-00059-f005:**
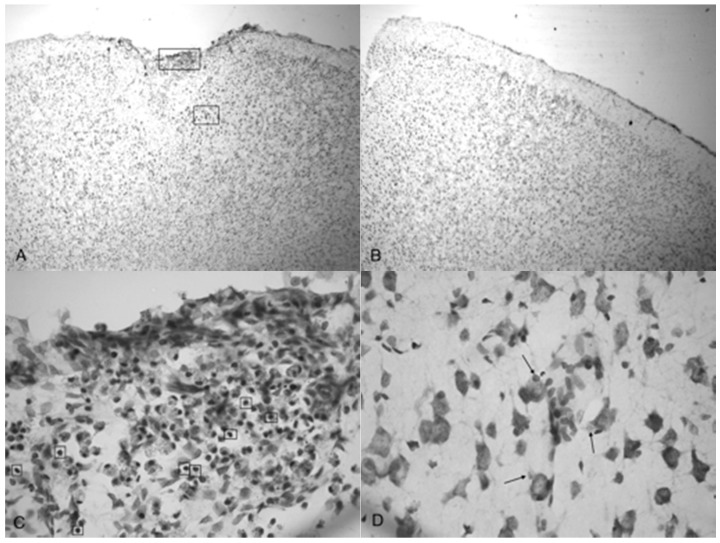
Photomicrographs taken of the frontal cortex stained with cresyl violet from two Han-Wistar rats, 10 days post-TBI from 5.0 bar acoustic wave. Image (**A**) shows the TBI-targeted left frontal cortex and Image (**B**) shows the contralateral intact right cortex at the same stereologic section. Indentation and damage of the cortex can be easily seen in the TBI-damaged hemisphere (40× magnification). Images (**C**,**D**) are higher magnification photos (100×) taken from the boxed regions ((**C**) from the upper box, and (**D**) from the lower box) of image (**A**). These images show the presence of condensed dark cells, neuroprotective astrocytes, and granulated cells, which are likely macrophages that have phagocytized blood products. The boxes in image (**C**) highlight the condensed dark cells, and the arrows in image (**D**) highlight granulated cells, likely macrophages that have phagocytized blood products.

**Figure 6 brainsci-07-00059-f006:**
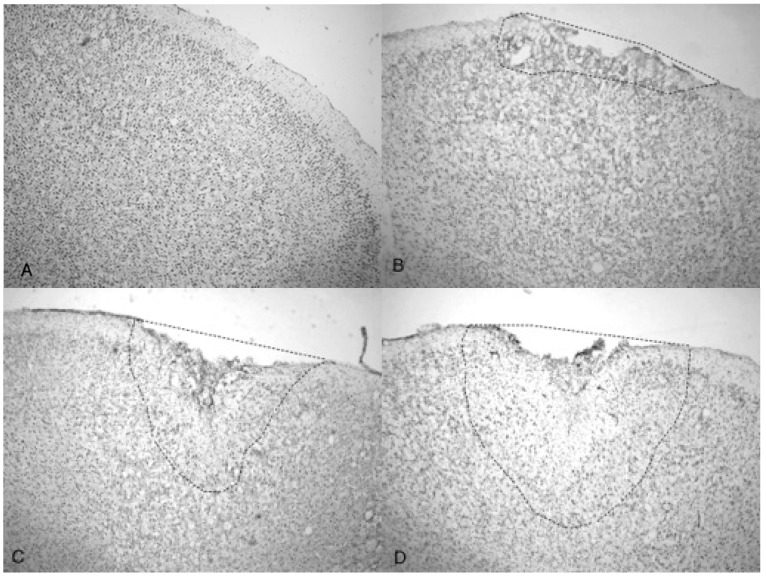
Photomicrographs taken of the frontal cortex showing the overall depth of injury and due to the penetration of acoustic waves: the control (0 bar), 3.4, 4.2 and 5.0 bar TBI, respectively of Han-Wistar rats, ten days post-injury. Image (**A**) shows no damage from an untreated, control rat (0 bar). Image (**B**) shows slight damage to the outer cortex (3.4 bar). Images (**C**,**D**) show the deeper penetrating impact delivered by the 4.2 and 5.0 bar. The dotted lines show examples of the infarct boundaries traced by locating the perimeter of condensed dark cells used to calculate the expanse of the area damaged. All images were taken at 40× magnification.

**Figure 7 brainsci-07-00059-f007:**
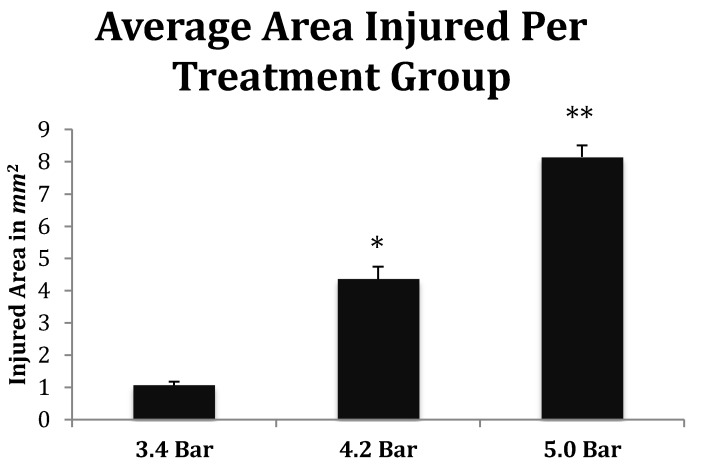
Quantitative measurements of the TBI injured area (mm^2^) from Han-Wistar rats (*n* = 10 for all treatments) subjected to acoustic wave treatments at 3.4, 4.2 and 5.0 bar. Quantification of the TBI infarct area was determined by the disruption of cells including the condensed dark cells which extend out to the periphery of the recovering border of the injury. The damaged cortical area surrounding the TBI was then traced by following the boundary of these peripheral cells. The infarct area was then quantified using SketchandCalc^TM^ to measure the region of damage (mm^2^). An ANOVA showed statistical differences among the treatment groups (*F* = 134.9; *p* < 0.001, denoted * and **) in a dose-dependent manner. All values are means ± standard error.

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
