# Peer review of "Validation of Acoustic Wave Induced Traumatic Brain Injury in Rats"

_brainsci, 2017, doi:10.3390/brainsci7060059_

Round 1

Reviewer 1 Report

The author established a new acoustic wave-induced closed traumatic brain injury (TBI) animal model in rat using Storz-D-11 Actor. They applied single pulse pressures to the rats and evaluated motor skills and memory in time sequential manner. They also evaluated histological changes at 10 days after TBI. Although some of the findings are new (immediately after acoustic TBI was administered to the left frontal cortex, every rat that received acoustic wave pressure above 3.4 bar TBI rolled and spun to its contralateral side (right) between one and four times before righting itself fully; This is known among researchers in baby rat model (with thin bone) but as far as reviewer understands, this phenomenon is not described in the literature), the author should discuss about the mechanism of the phenomenon they observed in the discussion. Also the reviewer strongly recommend to discuss with clinician familiar with TBI since the reviewer noticed multiple description is not medically correct.

L29. The mechanism of subdural hematoma is generally disruption of surface and bridging vein.

L31. The description is medically not correct.

L40. The description is medically not correct.

L44. The author should describe why they think civilian TBI have common aspect with that of acoustic induced TBI. See Cernak, I., and Noble-Haeusslein, L.J. (2009). Traumatic brain injury: an overview of pathobiology with emphasis on military populations. J. Cereb. Blood Flow Metab, and Chen, Y.C., Smith, D.H., and Meaney, D.F. (2009). In-vitro approaches for studying blast-induced traumatic brain injury. J. Neurotrauma 26, 861–876.

L70 The description is medical not correct. VS is definition in Glasgow Outcome Scale, not GCS.

L100 The author should describe whether they shave head or not. The presence of air theoretically makes difference in propagation of shock wave (acoustic waves as well) due to occurrence of impedance mismatch. The authors also should describe the reason for their choice of applied pressure (0, 3.4, 4.2 and 15 5.0 bar). Numbers of shock wave injury literatures suggest pressure dependence. See Delius, M. (2002). Twenty years of shock wave research at the Institute for Surgical Research. Eur. Surg. Res. 34, 30–36

L138 The authors should describe why they evaluated only 10 days after TBI. In the TBI literature as well as in the ischemia literature, generally they evaluate in time sequential manner to understand the time sequential change (usually 24 hrs, 72 hrs, at least and hopefully 7 days after the insults).

Fig 2 and 3. The authors should describe whether they observe pressure dependence effect of acoustic waves. The reviewer observe no apparent trend according to the pressure the rat received. Also, the author should describe why the result is not following pressure dependent manner at some point such as 7 days after TBI in Fig 2 and Fig. 3.

Fig 5 and 6. According to shock wave literature, the author should describe the findings that this is infarction instead of subacute findings after the occurrence of hemorrhage. See Nakagawa, A., Fujimura, M., Kato, K., Okuyama, H., Hashimoto, T., Takayama, K., and Tominaga, T. (2008). Shock wave-induced brain injury in rat: novel traumatic brain injury animal model. Acta Neurochir. Suppl. 102, 421–424 and Kato, K., Fujimura, M., Nakagawa, A., Saito, A., Ohki, T., Takayama, K., and Tominaga, T. (2007). Pressure-dependent effect of shock wave on rat brain: induction of neuronal apoptosis mediated by caspase-dependent pathway. J. Neurosurg.106, 667–676.

Author Response

We would like to thank the reviewer for their great insight into traumatic brain injury and for their careful and thoughtful inspection of this work. I have taken as many of their suggestions as I can and have incorporated them into our paper. I hope that they see their ideas and comments reflected in the revisions that have been made. 

A few of the comments were not addressed for particular reasons. I have copied a list of responses to each critique the reviewer made. Please see the list included below and attached revision with track changes. 

Thank you again for your thoughtful and careful response. 

Best, 

-Sean Berman 

L29- see corrected wording which differentiates between mild, moderate and severe TBI

L32- see corrected wording which differentiates between mild, moderate and severe TBI

L40- disagree with Reviewer 2’s comments based on:

Inglese, M., Makani, S., Johnson, G., Cohen, B.A., Silver, J.A., Gonen, O., Grossman, R.I., 2005. Diffuse axonal injury in mild traumatic brain injury: a diffusion tensor imaging study. J. Neurosurg. 103, 298303.

Kraus, M.F., Susmaras, T., Caughlin, B.P., Walker, C.J., Sweeney, J.A., Little, D.M., 2007. White matter integrity and cognition in chronic traumatic brain injury: a diffusion tensor imaging study. Brain 130, 25082519. 

Suh, M., Kolster, R., Sarkar, R., McCandliss, B., Ghajar, J., 2006. Cognitive and neurobiological research consortium. Deficits in predictive smooth pursuit after mild traumatic brain injury. Neurosci. Lett. 401, 108113.

 L44- see already included multiple references. Not sure how to adjust when widely supported. 

L70- omitted “vegetative state”

L100- did not shave head as all rats were raised the same and kept in same conditions. Had hair of similar length/thickness, and would have resulted in equal error throughout all groups in a all tests. Also, blast victims in real life generally have hair and is an important component to include. 

Decided that blast intensity needed to be greater than 3.0 based on pre testing trials using animal like models and high speed cameras. Anything lighter than 3.0 would not have been strong enough to make a detectible impact. 

L138 It’s well known that mild and moderate TBIs do not always resolve within 7 days and certainly moderate TBIs typically take 7 days or longer. We felt that 10 days was necessary to obtain more behavioral data, which indeed was significant as 4 out of 6 of our injured groups (3.4 and 4.2 bar Rotarod and 4.2 and 5.0 bar Morris Water Maze) actually regressed further from day 7 to day 10 when tested. This data proved significant. 

Figure 2 and 3- Both figures show that the blasts created deficits in motor skills ability and memory when compared to the sham. They are all significantly different, although these results do not appear to be directly correlated to the blast intensities. 

Figure 5 and 6- I do not understand the critique here and what the reviewer is asking for. 

Reviewer 2 Report

Overall I found this to be a very well written article although I would suggest a little more clarity when describing a TBI. Although I personally do not like animal experiments I appreciate the need and I feel that this article may possible allow better studies to be carried out on animals which may in turn lead to better treatment for all head trauma patients.

Author Response

We would like to thank the reviewer for their good insight, their appreciation and respect for animals (which we 100% share and only conduct animal research when we feel it is absolutely necessary to show a greater good for relevant clinical applications), and their positive review. 

-Sean Berman 

Round 2

Reviewer 1 Report

The authors presented an simple, reproducible animal model for pressure induced TBI.  Although more technical effort for understanding the mechanism of injury using this method should be clafified in the future, this pressure induced TBI animal model wouldl give some insight about pressure induced TBI and worth publication.